# Improvement in Electrochemical Performance of Waste Sugarcane Bagasse-Derived Carbon via Hybridization with SiO_2_ Nanospheres

**DOI:** 10.3390/molecules29071569

**Published:** 2024-03-31

**Authors:** Muhammad Mudassir Ahmad Alwi, Jyoti Singh, Arup Choudhury, SK Safdar Hossain, Akbar Niaz Butt

**Affiliations:** 1Department of Materials Engineering, College of Engineering, King Faisal University, P.O. Box 380, Al-Ahsa 31982, Saudi Arabia; malwi@kfu.edu.sa (M.M.A.A.); abutt@kfu.edu.sa (A.N.B.); 2Department of Chemical Engineering, Birla Institute of Technology, Ranchi 835215, India; jyotisingh161198@gmail.com; 3Department of Chemical Engineering, College of Engineering, King Faisal University, P.O. Box 380, Al-Ahsa 31982, Saudi Arabia

**Keywords:** waste sugarcane bagasse, activated carbon, SiO_2_ nanospheres, capacitance, supercapacitors

## Abstract

Sugar industries generate substantial quantities of waste biomass after the extraction of sugar water from sugarcane stems, while biomass-derived porous carbon has currently received huge research attention for its sustainable application in energy storage systems. Hence, we have investigated waste sugarcane bagasse (WSB) as a cheap and potential source of porous carbon for supercapacitors. The electrochemical capacitive performance of WSB-derived carbon was further enhanced through hybridization with silicon dioxide (SiO_2_) as a cost-effective pseudocapacitance material. Porous WSB-C/SiO_2_ nanocomposites were prepared via the in situ pyrolysis of tetraethyl orthosilicate (TEOS)-modified WSB biomass. The morphological analysis confirms the pyrolytic growth of SiO_2_ nanospheres on WSB-C. The electrochemical performance of WSB-C/SiO_2_ nanocomposites was optimized by varying the SiO_2_ content, using two different electrolytes. The capacitance of activated WSB-C was remarkably enhanced upon hybridization with SiO_2_, while the nanocomposite electrode demonstrated superior specific capacitance in 6 M KOH electrolyte compared to neutral Na_2_SO_4_ electrolyte. A maximum specific capacitance of 362.3 F/g at 0.25 A/g was achieved for the WSB-C/SiO_2_ 105 nanocomposite. The capacitance retention was slightly lower in nanocomposite electrodes (91.7–86.9%) than in pure WSB-C (97.4%) but still satisfactory. A symmetric WSB-C/SiO_2_ 105//WSB-C/SiO_2_ 105 supercapacitor was fabricated and achieved an energy density of 50.3 Wh kg^−1^ at a power density of 250 W kg^−1^, which is substantially higher than the WSB-C//WSB-C supercapacitor (22.1 Wh kg^−1^).

## 1. Introduction

Globally, fossil fuel consumption has led to an increase in energy demand and environmental sustainability [1,2,3]. Over the last few decades, a number of factors including population growth, fossil fuel exhaustion, and waste material production have contributed to the increasing demand for limitless energy sources and storage devices [4,5]. There is a growing demand for environmentally friendly, affordable, and renewable energy systems. To make these systems reliable, an efficient and cost-effective energy storage system is necessary for securing power at low cost [6,7]. Research and commercial interest are growing in the use of waste biomass to produce porous carbon-based electrode materials for cost-effective and efficient energy storage systems, since biomass-derived carbon offers low cost, sustainability, porosity, and excellent electrical conductivity [8]. This strategy can boost up the global economy as well as reduce the waste biomass-derived environmental pollution [9]. Furthermore, it will help to reduce the environmental impact of burning waste biomass for energy [10].

For supercapacitors, carbon is the most commonly used electrode material because it offers excellent electrical conductivity, chemical resistance, and high thermal and mechanical stability. Various carbon materials including carbon nanotubes, graphene, and activated carbon have been widely investigated as electrode materials for supercapacitors [11]. For supercapacitor manufacturers, low material costs are the most important issue, in addition to different technical issues. Recently, biomass-derived carbon has become increasingly popular for manufacturing supercapacitors, due to its low cost, abundance, and environmental friendliness [12]. However, the capacitive performance of these carbon materials is very much dependent on their physical properties such as surface area, mesoporosity, electrical conductivity, and electrochemical stability [13,14,15]. Researchers have explored various chemical and physical activation processes to manipulate the porosity and surface area of biomass-derived carbon materials and thus enhance their electrochemical capacitive performances [16]. However, the activation treatment can somewhat improve the specific capacitance and charge–discharge capacity of biomass-derived carbon, but it requires further improvement in the capacitance and energy density of biomass-derived carbon to meet commercial demand [17,18].

Hybridization of biomass-derived carbon with redox-active materials can enhance capacitance through the combined effects of electrical double-layer capacitance (EDLC) and pseudocapacitance [19,20,21]. Numerous pseudocapacitance materials including transition metal oxides/sulfides [22,23,24,25], conducting polymers [26,27], and transition metal MXenes [28,29,30] have been extensively investigated to enhance the electrochemical capacitance of carbon materials via hybridization. Nonetheless, there have been limited attempts to enhance the energy density of biomass-derived carbon by incorporating pseudocapacitance materials. In two studies, Wu et al. [16] and Hu et al. [31], bamboo-derived carbon was enhanced by hybridization with nickel hydroxide and copper oxide. Tang et al. [32] achieved a specific capacitance of 234.2 F/g at 1 A/g and a rate capability of 71.4% at 20 A/g for bamboo shoot shell-derived carbon/SiC composites. Recently, the banana peel-derived carbon/MnO_2_ composite demonstrated a specific capacitance of 139.6 F/g at 300 mA/g with 92.3% capacitance retention after 1000 cycles [33]. Recently, agro-waste biomass-derived carbon materials have gained increasing interest for supercapacitor applications because of their high surface area and tannable porosity [34,35,36,37,38,39,40,41,42,43,44,45,46]. Among various agro-wastes, sugarcane bagasse (*Sachharum officinarum*) is a typical waste biomass generated in huge quantities in the sugar industries. Most of this waste sugarcane bagasse is disposed of on land, causing serious environmental pollution. Hence, it is a great opportunity to investigate these waste bagasse fibers as a possible precursor for producing porous carbon and then to make use of this carbon as a supercapacitor electrode material. Earlier, researchers studied the electrochemical capacitive performance of SB-derived carbon and achieved low capacitance [47,48,49,50]. Recently, a few research groups tried to enhance the capacitance of SB-derived carbon by the incorporation of pseudocapacitance materials like metal oxides [51] and conducting polymers [52].

Indian sugar mills produce ~3 tons of waste sugarcane bagasse per annum, and each mill discards this huge waste bagasse as residue or uses it as a source of heat to meet the sugar mill’s heat demand. In the present investigation, we explored waste sugarcane bagasse as a cheap carbon source and SiO_2_ as an environmentally friendly and cheap pseudocapacitance material to prepare WSB-C/SiO_2_ nanocomposites with enhanced capacitive performance. The nanocomposites were prepared via the one-step pyrolytic decomposition of TEOS-modified bagasse fibers. The effects of SiO_2_ concentrations on the morphology and porosity of WSB-C/SiO_2_ nanocomposites were evaluated. The electrochemical measurements of different formulated WSB-C/SiO_2_ nanocomposites were studied on both alkaline and neutral electrolytes to compare their capacitive performance and electrochemical stability.

## 2. Results and Discussions

The functional groups generated in the different formulated waste SB-derived carbon materials were identified by FTIR spectroscopic analysis. Figure 1a illustrates the FTIR spectra of porous WSB-C and the different formulated WSB-C/SiO_2_ nanocomposites. Several IR peaks appear in the FTIR spectrum of porous WSB-C (Figure 1a). In addition, a strong peak at 1191 cm^−1^ and a weak peak at 907 cm^−1^ were observed for the stretching and bending vibration of C-C bond conjugated to C=C bond, respectively, while a medium IR band at 1643 cm^−1^ appeared for the stretching mode of C=C bonds [53]. The peak at 622 cm^−1^ raised in the spectra was perhaps due to the out-of-plane deformation of -OH groups [54]. In contrast to C-SB, the WSB-C/SiO_2_ nanocomposites revealed many IR peaks related to SiO_2_. The peaks at 1092 and 487 cm^−1^ can be attributed to the intrinsic vibration of Si-O-Si and Si-O bonds [30], indicating the pyrolytic formation SiO_2_ nanospheres from TEOS. During the carbonization process of TEOS-modified SB fibers, the -O-Si(OEt)_3_ groups were detached from cellulose chains by the pyrolytic dissociation of C-O bonds, which are weaker than Si-O bonds, followed by hydrolysis into orthosilicic acid (Si(OH)_4_) and finally thermal decomposition into SiO_2_ [55,56]. Raman spectra of WSB-C and the different WSB-C/SiO_2_ nanocomposites are displayed in Figure 1b. Two characteristic Raman active bands at around 1593 and 1356 cm^−1^ appeared for all nanocomposite samples. The low frequency G band is related to graphitic lattice vibration with E_2g_ mode, while the D band represents defects in graphitic structure. The degree of disorder in the different formulated samples was characterized by determining their I_D_/I_G_ ratio [57]. As shown in Figure 1b, the I_D_/I_G_ value of WSB-C is significantly lower than that of the WSB-C/SiO_2_ nanocomposites, indicating that there is a greater degree of graphitization in WSB-C as compared to the nanocomposites. The in situ pyrolytic growth of SiO_2_ nanospheres during carbonization of the TEOS-modified SB matrix perhaps impedes the graphitization kinetics. However, the higher I_D_/I_G_ values of the nanocomposites suggest the greater extent of an amorphous phase in the nanocomposites compared to WSB-C, which can facilitate their electrochemical capacitive performances. The X-ray diffraction patterns of the formulated WSB-C/SiO_2_ nanocomposites are illustrated in Figure 1c. The XRD pattern of WSB-C reveals a strong peak at 2θ = 26.4°, assigned to the (002) plane of the graphite lattice [58]. This high intense graphitic (002) peak indicates the formation of highly crystalline carbon as a result of the carbonization of SB-derived crystalline cellulose, which was prepared by alkaline treatment of raw SB prior to the carbonization process. In contrast, the WSB-C/SiO_2_ nanocomposites display several diffraction peaks at 2θ = 20.4°, 36.5°, 39.3°, 40.2°, 42.3°, and 50° corresponding to the (100), (110), (102), (111), (200), and (112) crystalline planes of SiO_2_, respectively [59], along with a (002) peak of crystalline graphite. The intensity of the graphitic (002) peak gradually decreases with increasing SiO_2_ concentration in the nanocomposites, which might be related to the degree of graphitization of the SB matrix. It can be noticed that the position of the (002) crystalline peak slightly shifted toward lower 2θ values for the WSB-C/SiO_2_ nanocomposites compared to WSB-C. The lower 2θ values of the WSB-C/SiO_2_ nanocomposites indicate their higher *d*-spacing compared to WSB-C. The *d*-spacing of WSB-C, WSB-C/SiO_2_ 1025, WSB-C/SiO_2_ 105, and WSB-C/SiO_2_ 11 are calculated to be 0.33, 0.341, 0.358, and 0.347 nm, respectively. The results indicate the disordered layer structure of graphite with multi-layered stacking [60], which is well in agreement with the Raman spectroscopy results. The greater *d*-spacing of the WSB-C/SiO_2_ nanocomposites can facilitate the easy diffusion of electrolyte ions into electrodes during electrolysis, making them potential candidates for energy storage applications. Furthermore, there is a weak and broad peak appearing in the 2θ range of 21–25° for the WSB-C/SiO_2_ nanocomposites, indicating some degree of amorphous SiO_2_ and carbon [61]. The surface characteristics of SB-derived carbon and its nanocomposites were evaluated by BET analysis. The N_2_ adsorption–desorption isotherms of WSB-C and the formulated WSB-C/SiO_2_ nanocomposites are illustrated in Figure 2a. A mixture of type I and type IV isotherms is observed for both WSB-C and the WSB-C/SiO_2_ nanocomposites, indicating a typical combination of micropores and mesopores [62]. The initial part of the isotherm follows type I adsorption, which is attributed to the controlled monolayer adsorption by micropores [63]. However, the middle of the isotherm shows a characteristic hysteresis loop indicating the presence of large mesopores with a wide pore size distribution. The specific surface areas and total pore volumes of the different samples are presented in Figure 1. The specific surface areas and pore volumes were decreased for the WSB-C/SiO_2_ nanocomposites compared to WSB-C. The maximum S_BET_ and total pore volume of 342 m^2^/g and 0.0426 cm^3^/g were recorded for WSB-C, while the lowest values of 115.7 m^2^/g and 0.0219 cm^3^/g were obtained for the WSB-C/SiO_2_ 11 nanocomposite containing higher concentrations of SiO_2_ nanoparticles. This could be attributed to the blocking of micropores by SiO_2_ nanoparticles, as evidenced in the FE-SEM images. In contrast to microporosity, the mesoporosity of the WSB-C/SiO_2_ nanocomposites is significantly increased compared to WSB-C, and the C/SiO_2_ 105 nanocomposite exhibits the highest amount of mesoporosity with 74.2%. The BJH pore size distributions of WSB-C and the different formulated WSB-C/SiO_2_ nanocomposites are displayed in Figure 2b. The average diameters of the pores of the different samples are presented in Table 1. The pore size is gradually increased with increasing SiO_2_ content, which might be the result of pore enlargement caused by heterogeneous shrinkage during the carbonization process [64].

The surface chemical compositions of the as-synthesized WSB-C/SiO_2_ nanocomposites were determined by XPS analysis. Figure 3a demonstrates the XPS survey curves of activated SB-derived carbon and the different WSB-C/SiO_2_ nanocomposites. The XPS signals for three major elements such as carbon (C), oxygen (O), and silicon (Si) at ~285 eV, ~531 eV, and ~104 eV are observed in the survey curves of the WSB-C/SiO_2_ nanocomposites, while WSB-C contained no elemental silicon. As is obvious, the peak intensity of elemental Si for the nanocomposites gradually increased with increasing TEOS loading in the modified-SB fibers. Figure 3b displays the core level Si 2p spectra of WSB-C/SiO_2_ 105 consisting of a single Gaussian peak at 104.3 eV, corresponding to the Si-O-Si bond of SiO_2_ nanospheres [62]. The C 1s spectra is deconvoluted into four peaks at 284.1 eV, 284.9 eV, and 287.5 eV, shown in Figure 3c, which are associated with C=C, C-C, and C=O bonds, respectively. The atomic percentages of elemental C, Si, and O in the different formulated carbon samples are demonstrated in Figure 3d. The results indicate that the atomic percentage of carbon is reduced for the WSB-C/SiO_2_ nanocomposites compared to WSB-C. The lowest yield of 77.21% was recorded for WSB-C/SiO_2_ 11, which is 17% lower than that achieved for WSB-C. Furthermore, the at% of carbon is gradually decreased with increasing TEOS content in modified-SB. These findings suggest that the degree of carbonization/graphitization of SB was affected by the in situ formation of SiO_2_, which might be due to the change in the heat of carbonization during the pyrolysis process.

The microstructures of SB-derived carbon and its hybrids with SiO_2_ were observed with SEM analysis and the resulting images are shown in Figure 4. The orderly aligned channels appear in the cross-sectional images of the nanocomposites with multiple sizes of pores on their surfaces. The top surface of the WSB-C displays slightly crumpled features with multiple size pores (Figure 4a). In contrast, the WSB-C/SiO_2_ nanocomposites clearly exhibit a uniform distribution of SiO_2_ nanospheres on the porous carbon surface. The concentration of SiO_2_ nanospheres gradually increases with increasing TEOS loading in the modified-SB precursor. For the WSB-C/SiO_2_ 1025 and WSB-C/SiO_2_ 105 nanocomposites, the majority of SiO_2_ nanospheres remain discrete and embedded into the carbon matrix (Figure 4d), while the SiO_2_ nanospheres are largely agglomerated in the WSB-C/SiO_2_ 11 nanocomposite (Figure 4e). The extensive agglomeration of SiO_2_ nanospheres reduced the specific surface area and limited the pseudocapacitive effects of the WSB-C/SiO_2_ 11 nanocomposite, which have been observed in the BET and CV results. The cross-sectional view of the WSB-C/SiO_2_ nanocomposites in Figure 4f–h exhibits three-dimensional (3D) channels within the carbon matrix. This 3D structure provides plenty of space for growing SiO_2_ nanospheres. Like tree trunks, these channels remain aligned along their longitudinal axis, which can facilitate the transport of electrolyte through them. Furthermore, the cross-sectional images reveal a uniform dispersion of SiO_2_ nanospheres throughout the channel wall, which can facilitate the access of electrolyte ions to redox-active SiO_2_ and thus enhance the capacitance efficiency of the nanocomposites.

The effects of SiO_2_ nanospheres on the electrochemical properties of SB-derived activated carbon were evaluated using CV and GCD measurements. Figure 5a demonstrates the CV profiles of WSB-C and the different WSB-C/SiO_2_ nanocomposites in a potential range of -1.0 to 0 V at 50 mV/s. The WSB-C electrode reveals a nearly rectangular CV curve with no obvious redox peaks, suggesting the behavior of an electrical double-layer capacitor with fast charge and discharge processes. On the other hand, the CV curves of the WSB-C/SiO_2_ nanocomposite electrodes reveal a distorted rectangular shape with a pair of faradaic redox peaks, suggesting both EDLC and pseudocapacitance characteristics of the WSB-C/SiO_2_ electrode. The oxidation and reduction peaks appear at −0.12 and −0.30 V, respectively. The redox reactions involved during the electrochemical process in the KOH electrolyte can be illustrated as follows [65]:(1)$${Si}^{II}O_{2}+K^{+}+e^{-}↔{Si}^{III}OOK\;$$

For the WSB-C/SiO_2_ electrodes, the anodic peak current gradually increased for the WSB-C/SiO_2_ 1025 and WSB-C/SiO_2_ 105 electrodes but it decreased for the WSB-C/SiO_2_ 11 electrode. The WSB-C/SiO_2_ 105 electrode exhibited better pseudocapacitive characteristics in terms of current response to voltage. However, the decrease in anodic current of the WSB-C/SiO_2_ 11 electrode was the result of the agglomeration of SiO_2_ nanospheres (as shown in the FE-SEM image) and the subsequent decrease in surface area of active SiO_2_. The CV curves of the WSB-C electrode in a wide range of scan rates are displayed in Figure 5b. The CV curves retained a nearly rectangular shape even at a high scan rate of 125 mV/s, indicating the fast and reversible EDLC behavior of the WSB-C electrode. The shape of the CV profiles of the WSB-C/SiO_2_ 105 electrode remained unchanged with a small shift in redox peak position upon increasing the scan rate (Figure 5c), suggesting its high-rate capability over a wide range of scan rates.

It is necessary to study the kinetics of charge storage in WSB-C/SiO_2_ electrodes at different potentials in order to identify the mechanism behind the charge stored in them. In general, there are two different mechanisms through which the total charge is stored: (i) diffusion mechanism (intercalation/deintercalation) and (ii) capacitive mechanism [64]. The Power’s law was used to characterize this phenomenon [65]:(2)$$i\;\left(V\right)=a\vartheta^{b}\;$$
where ϑ is the scan rate and *b* is the slope of the linear plot of Log (*i*) vs. Log (ϑ). Two conditions based on the *b* values determine the mechanism responsible for charge storage: if *b* = 0.5, the diffusion-controlled intercalation process is dominant over the capacitive process, and when *b* = 1.0, the capacitive contribution is higher than the diffusion contribution. Figure 5d demonstrates the logarithmic linear plots between redox current and scan rate. In the present case, the *b* values for anodic and cathodic processes are 0.6835 and 0.7386, respectively, which suggests that the charge storage process is governed by both surface-controlled and diffusion-controlled mechanisms. Furthermore, the diffusion-controlled faradaic process made a considerable contribution to enhancing the charge storage capability of the WSB-C/SiO_2_ 105 electrode. The large fraction of mesopores in the WSB-C/SiO_2_ 105 nanocomposite facilitated the intercalation/deintercalation of electrolyte ions. The relative currents from the capacitive and diffusion processes at different scan rates can be determined using the given equation [66]:(3)$$i\;\left(V\right)=k_{1}\left(\vartheta \right)+k_{2}\vartheta^{1/2}\;$$
where $k_{1}\left(\vartheta \right)$ and $k_{2}\vartheta^{1/2}$ values correspond to the current contribution from the diffusion and capacitive processes, respectively. Figure 5e reveals the percent contribution of the diffusion and capacitive processes to the CV area at 5 mV/s. The low scan rate favors the diffusion process and thus diffusion is dominant over the capacitive contribution. As scan rates increase, the diffusion contribution gradually decreases, and the capacitive contribution steadily increases (Figure 5f).

The impedance analyses (EIS) were further carried out to investigate the ionic diffusion kinetics for the as-synthesized WSB-C and its nanocomposites with SiO_2_ [67]. The Nyquist plots, shown in Figure 6a, allow us to compare the impedance properties of the different formulated electrode materials. WSB-C exhibits a solution resistance (*R_s_*) of 2.86 Ω due to the relatively low hydrophilicity of carbon material, while the low *R_s_* values of the WSB-C/SiO_2_ nanocomposites suggest their better hydrophilic characteristics. At high frequencies, Nyquist plots represent charge-transfer resistance (*R_CT_*), while at low frequencies a line with a 45° slope represents capacitive behavior. Table 2 presents the fitted values for each component for each material. The charge-transfer resistance of WSB-C was significantly reduced upon the inclusion of the SiO_2_ nanospheres. The lower charge-transfer resistance of the WSB-C/SiO_2_ electrodes reflects their electrochemical performance. The equivalent circuit model for WSB-C/SiO_2_ 105 is shown in the inset of Figure 6a.

The gravimetric charge–discharge (GCD) profiles of the pure WSB-C and the different WSB-C/SiO_2_ nanocomposite electrodes at a current density of 1 A/g are demonstrated in Figure 6b. A nearly symmetric triangular charge–discharge curve appears for the WSB-C electrode, indicating pure non-faradaic EDLC behavior. The WSB-C electrode reveals a Coulombic efficiency of 100.6% at 1 A/g, suggesting its excellent electrochemical reversibility. In contrast, the WSB-C/SiO_2_ nanocomposite electrodes display asymmetric distorted triangular charge–discharge curves with an obvious discharge plateau at ~0.4 V, suggesting faradaic contribution to the overall charge storage process. Furthermore, there was a longer discharge time for the WSB-C/SiO_2_ electrodes than for the WSB-C electrodes, indicating that the nanocomposites have superior capacity for storing charge than the WSB-C electrodes. Among the nanocomposite electrodes, the WSB-C/SiO_2_ 105 electrode took the maximum time to discharge, indicating greater charge storage capability. Figure 6c illustrates the GCD curves of the WSB-C electrode at different current densities. The symmetric features of the GCD curves remain unaffected upon increasing the current density, indicating excellent rate capability. The GCD profiles of the WSB-C/SiO_2_ 105 nanocomposite electrode at various current densities between 0.25 and 5 A/g are demonstrated in Figure 6d. The asymmetric features of the GCD curves appear for all GCD curves due to the pseudocapacitance effects. The gravimetric specific capacitances (*C_gsp_*) of the WSB-C and the WSB-C/SiO_2_ nanocomposite electrodes were calculated from their respective GCD curves, using Equation (4):(4)$$C_{gsp}=\frac{I∆t}{m∆V}$$

Figure 6e demonstrates the variation in specific capacitances as a function of current densities for the different electrodes. The results clearly exhibit a significant improvement in the specific capacitance of SB-derived carbon upon hybridization with SiO_2_. The specific capacitances of the WSB-C/SiO_2_ 1025, WSB-C/SiO_2_ 105, and WSB-C/SiO_2_ 11 electrodes are ~160%, ~240%, and ~210% higher than those of the pure WSB-C electrodes, respectively. The highest specific capacitance of 362.3 F/g at 0.25 A/g was achieved for the WSB-C/SiO_2_ 105 electrode. Table 3 presents that the specific capacitance of the WSB-C/SiO_2_ 105 electrode is higher than those previously reported for similar types of electrode materials. The capacitance values gradually reduced from 362.3 to 220.6 F/g when the current density increased from 0.25 to 5 A/g, indicating an excellent capacitance retention of 61% even with a 20-fold increase in current density. However, a relative lower capacitance retention was observed for WSB-C/SiO_2_ 1025 (49% retention) and WSB-C/SiO_2_ 11 (37% retention). The lower capacitance retention in the WSB-C/SiO_2_ 1025 and WSB-C/SiO_2_ 11 electrodes might be ascribed to the presence of a smaller fraction of mesopores and some fraction of unused SiO_2_ at high current densities. The cycling stability of the WSB-C/SiO_2_ nanocomposite electrodes was compared with that of the WSB-C electrode, shown in Figure 6f. The nanocomposite electrodes revealed somewhat lower cycling stability than the WSB-C electrode, which might be due to their limited pseudocapacitance contribution at higher cycles [68]. The highest cycling stability of 97.4% was achieved for the WSB-C electrode, while the cycling stability went down to 91.7% and 86.9% for the WSB-C/SiO_2_ 105 and WSB-C/SiO_2_ 11 electrodes, respectively. This might be due to the relatively low electrochemical stability of metal oxide compared to carbon.

The capacitive performance of the WSB-C/SiO_2_ 105 electrode was further evaluated in neutral electrolyte, i.e., 1 M Na_2_SO_4_, and the results were compared with those obtained in alkaline electrolyte, i.e., 6 M KOH. The CV profiles of the WSB-C/SiO_2_ 105 electrode over the potential window of −1.0 to 0 V in 1 M Na_2_SO_4_ electrolyte at different scan rates are illustrated in Figure 7a. Nearly rectangular-shaped CV curves with a pair of oxidation and reduction peaks centered at −0.49 V and −0.64 V are observed, indicating the involvement of both EDLC and pseudocapacitance mechanisms in the charge storage process. The redox peaks are associated with the reactions given in Equation (5) [76]:(5)$${Si}^{II}O_{2}+{Na}^{+}+e^{-}↔{Si}^{III}OONa\;$$

There is almost no change in the shape of the CV profiles when the scan rate is increased from 5 to 125 mV/s, indicating the high-rate capability and electrochemical stability of the nanocomposite electrode in Na_2_SO_4_ electrolyte. The total current gradually increases with the increasing scan rate. The GCD profiles for the two-electrode system at various current densities are displayed in Figure 7b. The distorted triangular shape of the GCD curves with a voltage plateau further suggests the occurrence of faradaic surface-redox reactions during the charge storage process. The Coulombic efficiency was determined to be 100.9%. The specific capacitances at different current densities were determined from the corresponding GCD curves, using Equation (4). Figure 7c reveals the gravimetric specific capacitance of the WSB-C/SiO_2_ 105 electrode at different current densities in two different electrolytes, i.e., 6 M KOH and 1 M Na_2_SO_4_. The maximum specific capacitance of 319.8 F/g at 0.25 A/g was achieved in 1 M Na_2_SO_4_ electrolyte. As can be seen in Figure 7c, the capacitive performance of the nanocomposite electrode is somewhat lower in the Na_2_SO_4_ electrolyte than in the KOH electrolyte. Compared to K^+^ ions, Na^+^ ions have a bigger size and higher internal resistance, resulting in a slower transport rate, which results in lower capacitance. With the Na_2_SO_4_ electrolyte, the WSB-C/SiO_2_ 105 electrodes show a lower rate capability of 46.8% than with the KOH electrolytes (61%). For hydrated Na^+^ ions, their larger size is indeed unfavorable to smooth transportation and diffusion, especially at higher current densities, resulting in lower capacitance. The nanocomposite electrode exhibited superior cycling stability with capacitance retention of 95.6% after 10,000 cycles in 1 M Na_2_SO_4_ compared to that in 6 M KOH (91.7%), as shown in Figure 7d. The relatively lower cycling stability in the alkaline electrolyte might be due to the harsh and corrosive nature of the strong alkaline electrolyte (KOH) compared to the neutral Na_2_SO_4_ electrolyte. Figure 7e demonstrates the Ragone plots for the as-assembled symmetric WSB-C//WSB-C and WSB-C/SiO_2_ 105//WSB-C/SiO_2_ 105 devices in KOH and Na_2_SO_4_ electrolytes. The WSB-C/SiO_2_ 105//WSB-C/SiO_2_ 105 device in 6 M KOH electrolyte delivered a maximum energy density of 50.3 WH kg^−1^ at a power density of 250 W kg^−1^, which is significantly greater than that produced by the WSB-C//WSB-C device (22.1 Wh kg^−1^ at 250 W kg^−1^). The as-achieved energy density is superior to earlier reported biomass-derived carbon-based symmetric supercapacitors such as silkworm cocoon-derived carbon (34.4 Wh kg^−1^) [77]; wheat bran-derived carbon (32.7 Wh kg^−1^) [78]; peanut meal-derived carbon (24.9 Wh kg^−1^) [79]; rice straw-derived carbon (7.8 Wh kg^−1^) [80]; coconut shell-derived carbon (14.7 Wh kg^−1^) [81]; sugarcane bagasse-derived carbon (37.5 Wh kg^−1^) [82]; and cornstalk-derived carbon (10 Wh kg^−1^) [83]. Furthermore, the SC device achieved higher energy density in KOH electrolyte than in Na_2_SO_4_ electrolyte (44.4 Wh kg^−1^). The mechanism of electrochemical process and LED lightening of the as-fabricated symmetric WSB-C/SiO_2_ 105//WSB-C/SiO_2_ 105 device is displayed in Figure 1. The LED bulb (1.5 V) was lit with two symmetric WSB-C/SiO_2_ 105//WSB-C/SiO_2_ 105 electrodes after charging to 1.5 V, where the dimension of each electrode used in the device was 1.5 × 1.5 cm^2^. The LED bulb showed stable brightness for 30 min. The results indicate that the present WSB-C/SiO_2_//WSB-C/SiO_2_ symmetric supercapacitors have great potential for energy storage applications with impressive market value. This impressive energy density of waste sugarcane bagasse-derived carbon/SiO_2_ 105 makes it a potential cost-effective and environmentally friendly electrode material for next-generation supercapacitors.

## 3. Experimental

### 3.1. Materials

Waste sugarcane bagasse fibers were collected from a sugar mill in Uttar Pradesh, India. Potassium hydroxide (KOH), glacial acetic acid, triethyl orthosilicate, ethanol, and liquid ammonia solution were collected from Alfa Asare, India. All chemicals were utilized without undergoing additional purification.

### 3.2. Activation and Chemical Modification of Sugarcane Bagasse Fibers

The as-collected WSB fibers were initially washed thoroughly with DI water and dried at 60 °C. The dried WSB fibers were milled to size 40 mesh. For the activation process, 100 g WSB powder was dispersed in 500 mL 6 M KOH solution under stirring for 2 h. Afterward, the WSB fibers were treated with glacial acetic acid and washed with DI water to neutralize (pH = 7) them. The activated WSB fibers were then chemically modified with TEOS, where activated WSB fibers were mixed with different concentrations of TEOS (W_SB fiber_/W_TEOS_ = 1/0.25, 1/0.5, and 1/1) in ethanol/water (80/20 *v*/*v*) and stirred for 3 h. Liquid NH3 was drop-wisely added to the solution for neutralization. The resulting TEOS-modified WSB fibers were washed with DI water and thermally cured at 120 °C for 3 h.

### 3.3. Carbonization of Activated SB and TEOS-Modified SB Fibers

The alkaline-activated WSB and TEOS-modified WSB fibers were carbonized at 600 °C under nitrogen flow (20 mL/min) using a tube furnace. The carbonization of activated WSB and TEOS-modified WSB fibers produced porous carbon (WSB-C) and carbon/SiO_2_ nanocomposites, respectively. The nanocomposites obtained from TEOS-modified WSB fibers with fiber/TEOS ratios of 1/0.25, 1/0.5, and 1/1 are designated as WSB-C/SiO_2_ 1025, WSB-C/SiO_2_ 105, and WSB-C/SiO_2_ 11, respectively.

### 3.4. Structural Characterizations

FTIR and Raman analyses were conducted on Shimadzu IR-Prestige 21 (Shimadzu Corp, Japan) and Renishaw Raman System 3000 spectrophotometers (Renishaw, UK), respectively, using powdered samples. The XRD patterns of the samples were obtained with a Rigaku SmartLab diffractometer (Rigaku Corporation, Japan). The surface characteristics of the as-prepared materials were evaluated using a Micromeritics ASAP 2020 adsorption analyzer (Micromeritics Instrument Corporation, USA). An ESCA MultiLab 2000 spectroscopic analyzer (VG Systems Ltd. UK) was used to characterize the surface chemical compositions of the materials. A field-emission scanning electron microscope (FE-SEM, S-4700, Hitachi, Japan) was used to take microstructural images of the as-prepared samples.

### 3.5. Electrochemical Measurements

The electrochemical measurements were carried out in aqueous 6 M KOH electrolyte at room temperature (25 °C) using 3-electrode and 2-electrode cells for cycling voltametric (CV) and charge–discharge experiments, respectively. A potentiostat–galvanostat (SQUIDSTAT SOLO, Admiral Instruments, USA) was used for these measurements. For the 3-electrode cell, powdery sample-coated GCE with 2 µL Nafion solution (5 wt%), platinum sheet, and calomel electrode served as working, counter, and reference electrodes, respectively. For the 2-electrode configuration, two symmetric electrodes (1 × 1 cm^2^) consisting of active material-loaded Ni-foam were superimposed with a separator. The electrode was prepared by dispersing the WSB-C or WSB-C/SiO_2_ powder (~1.8 mg) on nickel foam and pressing under pressure of 15 tons, using a hydraulic press. EIS measurements were performed in a range of 100,000–0.1 Hz and an AC amplitude of 5 mV.

## 4. Conclusions

In summary, waste sugarcane bagasse-derived carbon/SiO_2_ nanocomposites were prepared through carbonization of TEOS-modified activated bagasse fibers. The capacitive performance of WSB-derived activated carbon was remarkably enhanced upon hybridization with SiO_2_, indicating a significant pseudocapacitive contribution from redox-active SiO_2_. A maximum capacitance of 362.3 F/g at 0.25 A/g was achieved for the WSB-C/SiO_2_ 105 nanocomposite, which is around 2.5 times higher than that of the WSB-derived activated carbon. However, higher SiO_2_ concentrations decrease the capacitance of the nanocomposite due to the agglomeration of SiO_2_ particles, as shown for WSB-C/SiO_2_ 11. The nanocomposite electrode demonstrated superior capacitance and energy density in KOH electrolyte compared to neutral Na_2_SO_4_ electrolyte. The WSB-C/SiO_2_ 105//WSB-C/SiO_2_ 105 supercapacitor achieved an energy density of 50.3 Wh kg^−1^ at a power density of 250 W kg^−1^ in 6 M KOH electrolyte, which is considerably higher than that achieved in 1 M Na_2_SO_4_ (44.4 Wh kg^−1^). However, the cycling stability of the supercapacitor was significantly higher in neutral Na_2_SO_4_ electrolyte than in alkaline electrolyte. Hence, the WSB-C/SiO_2_ 105 nanocomposite as an efficient, sustainable, and cheap electrode material could potentially be used in next-generation energy storage devices.

## Data Availability

Data are contained within the article.

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
