# Peer review of "Improvement in Electrochemical Performance of Waste Sugarcane Bagasse-Derived Carbon via Hybridization with SiO2 Nanospheres"

_molecules, 2024, doi:10.3390/molecules29071569_

Round 1

Reviewer 1 Report

Comments and Suggestions for Authors

This work used waste sugarcane bagasse to produce cheap carbon material, and investigate the electrochemical performance of WSB-C/SiO2 nanocomposites. This work is interesting, which offers an alternative route to prepare the carbon material used as electrode materials using waste sugarcane bagasse. Some suggestions are given below:

(1) Comparing the electrochemical performance of WSB-C//WSB-C supercapacitor with published works.

(2) Providing glowed LED indicator in order to justify the practical application of the reported WSB-C//WSB-C supercapacitor.

(3) Updating the references, for example,

* Journal of Energy Storage, 2023, 73(Part D): 109182.

* Journal of Advanced Ceramics, 2022, 11 (11): 1735-1750.

* Journal of Advanced Ceramics, 2022, 11 (5): 742-753.

* Advanced Ceramics, 2022, 43: 300-303.

Author Response

Com 1. Comparing the electrochemical performance of WSB-C//WSB-C supercapacitor with published works.

Ans. As suggested by reviewer, we have included the energy densities of various biomass-derived carbon-based symmetric supercapacitors reported earlier in the revised MSS for comparison with presently achieved energy density for symmetric WSB-C/SiO2 //WSB-C/SiO2 supercapacitor.

Com2. Providing glowed LED indicator in order to justify the practicalapplication of the reported WSB-C//WSB-C supercapacitor.

Ans. As suggested by reviewer, we have incorporated the details about the symmetric device and working performance of the LED bulb in the revised MSS. The results indicate the great potential of the present symmetric supercapacitor device in energy storage applications for several portable electronic devices.

Com3. 

Updating the references, for example,

* Journal of Energy Storage, 2023, 73(Part D): 109182.

*Journal of Advanced Ceramics, 2022, 11 (11): 1735-1750.

* Journal of Advanced Ceramics, 2022, 11 (5): 742-753.

  • Advanced Ceramics, 2022, 43: 300-303.

Ans. As suggested by reviewer, we have included all these references (2, 23, 25) in the revised MSS. The detail about the last reference may be wrong and consequently not included.  

Reviewer 2 Report

Comments and Suggestions for Authors

This work is a decent piece of work. Authors have discussed their results well. While the novelty is not sufficiently high, however, the work could be suitable for publication after this minor revision.

 1. Introduction and Conclusions are repetitive; both should be shortened accordingly.

2. Comparison is very shallow. Results should be compared with recent stste-of-the-arts including the following papers. For e.g., Advanced Functional Materials 34 (1), 2306815, DOI: 10.1002/adfm.202306815; Small Methods, DOI: 10.1002/smtd.202201551;  Energy & Environmental Materials e12516, DOI: 10.1002/eem2.12516; ACS Sustainable Chemistry & Engineering, DOI: 10.1021/acssuschemeng.8b01845;

3. The novelty should be highlighted in the beginning.

4. Authors should comment on the coulombic efficiency as noted in Figure 7.

5. Energy and Power density values calculations should be reported only for 2-electrode and devices data, not for 3-electrode configuration.

6. Language should be improved throughout the manuscript.

Comments on the Quality of English Language

Some grammatical errors

Author Response

Com. Introduction and Conclusions are repetitive; both should be shortened accordingly.

Ans. As suggested by reviewer, we have thoroughly revised both introduction and conclusion parts, considerably shorten. 

Com2. Comparison is very shallow. Results should be compared with recent state-of-the-arts including the following papers. For e.g., Advanced Functional Materials 34 (1), 2306815, DOI:10.1002/adfm.202306815; Small Methods, DOI:10.1002/smtd.202201551; Energy & Environmental Materialse12516, DOI: 10.1002/eem2.12516; ACS Sustainable Chemistry &Engineering, DOI: 10.1021/acssuschemeng.8b01845;

Ans. We have included all suggested references (22, 28, 29, 30) in the revised MSS.

Com3. The novelty should be highlighted in the beginning.

Ans. We have made the necessary modification in revised MSS.

Com4. Authors should comment on the coulombic efficiency as noted in Figure 7.

Ans. We have incorporated the coulombic efficiency in Na2SO4 electrolyte in the revised MSS.

Com5. Energy and Power density values calculations should be reported only for 2-electrode and devices data, not for 3-electrode configuration.

Ans. We agree with you. In the present study, the energy and power densities were determined for two electrode symmetric devices such as WSB-C//WSB-C and WSB-C/SiO2 105//WSB-C/SiO2 105

Com6. Language should be improved throughout the manuscript.

Ans. We have carefully removed the grammatical errors from revised MSS.

Round 2

Reviewer 1 Report

Comments and Suggestions for Authors

The revised manuscript can be accepted for publication.